# Migraine and gastroesophageal reflux disease: Disentangling the complex connection with depression as a mediator

**Zixiong Shen**[1], **Yewen Bian**[2], **Yao Huang**[3], **Wenhua Zhou**[4], **Hao Chen**[5], **Xia Zhou**[4]*, **Liuying Li**[4]*

1 Department of Thoracic Surgery, The First Hospital of Jilin University, Changchun, Jilin, China,
2 Department of Acupuncture and Physiotherapy, Nantong Third People's Hospital (Affiliated Nantong Hospital 3 of Nantong University), Nantong, Jiangsu, China, 3 Cardiovascular Center, The First Hospital of Jilin University, Changchun, Jilin, China, 4 Department of Traditional Chinese Medicine, Zigong First People's Hospital, Zigong, Sichuan, China, 5 Department of Neurosurgery, The First Hospital of Jilin University, Changchun, Jilin, China

* zhoux1823@163.com (XZ); arenally@sina.com (LL)

**Data Availability Statement:** Data on GERD and migraine (validation) can be downloaded from "https://www.ebi.ac.uk/gwas" with IDs "GCST90000514" and "GCST90000016". Data on migraine (discovery) and depression are sourced

## Abstract

### Objective

Gastroesophageal reflux disease (GERD) and migraine are public health concerns worldwide. No observational study has conclusively elucidated the causal relationship between these two conditions. We employed Mendelian randomization (MR) methods to explore the potential causal links between GERD and migraine.

### Methods

Genome-wide association studies were subjected to MR to infer the causality between GERD and migraine. Bidirectional two-sample MR was performed to establish causal relationships. Multivariable MR analysis was conducted to adjust potential confounding factors, and mediation MR analysis was utilized to assess the role of depression between GERD and migraine as a mediator. We primarily utilized the inverse variance weighted method (IVW) and sensitivity analysis methods, including MR-Egger, weighted median, and leave-one-out methods. We assessed heterogeneity and pleiotropy to ensure the reliability of the results.

### Results

Bidirectional two-sample MR revealed a positive causal effect of GERD on migraine (IVW: OR = 1.49, 95% CI: 1.34–1.66, p = 3.70E-13). Migraine did not increase the risk of GERD (IVW: OR = 1.07, 95% CI: 0.98–1.17, p = 0.1139). Multivariable MR indicated that the positive causal effect of GERD on migraine remained after adjustment for factors, such as smoking, alcohol consumption, obesity, type 2 diabetes, and depression. Mediation MR revealed that depression mediated 28.72% of GERD's effect on migraine. MR analysis was supported by all sensitivity analyses and was replicated and validated in another independent dataset on migraine.

from FinnGen ("https://www.finngen.fi/fi"). Data on smoking and alcohol consumption are obtained from the GWAS & Sequencing Consortium of Alcohol and Nicotine use (GSCAN) ("https://genome.psych.umn.edu/index.php/GSCAN"). Data on Body Mass Index (BMI) are sourced from the Genetic Investigation of Anthropometric Traits (GIANT) ("https://portals.broadinstitute.org/collaboration/giant"). Data on Type 2 Diabetes (T2D) is obtained from the Diabetes Genetics Replication and Meta-analysis (DIAGRAM) ("http://www.diagram-consortium.org/").

**Funding:** This study was funded by The Key Research and Development Plan (Major Science and Technology Special Project) of the Sichuan Provincial Science and Technology Department (Number: 2022YFS0392), the Key Discipline Construction Project of the Sichuan Provincial Administration of Traditional Chinese Medicine (Number: 202072), and 2023 Sichuan Provincial Administration of Traditional Chinese Medicine (2023MS252). There was no additional external funding received for this study. The funder had no role in the study design, data collection and analysis, decision to publish, or preparation of the manuscript.

**Competing interests:** The authors have declared that no competing interests exist.

**Abbreviations:** GERD, gastroesophageal reflux disease; RCT, randomized controlled trial; IV, instrumental variable; MR, Mendelian randomization; TSMR, two-sample Mendelian randomization; MVMR, multivariable Mendelian randomization; MMR, mediation Mendelian randomization; GWAS, Genome-wide association studies; UKBB, UK Biobank; GSCAN, GWAS & Sequencing Consortium of Alcohol and Nicotine use; CigDay, daily cigarette consumption; DrnkWk, weekly alcohol consumption; BMI, body mass index; GIANT, Genetic Investigation of Anthropometric Traits; T2D, Type 2 diabetes; DIAGRAM, Diabetes Genetics Replication and Meta-analysis; SNP, single nucleotide polymorphism; IVW, inverse variance weighting; MR-Egger, MR-Egger regression; MR-PRESSO, Mendelian Randomization Pleiotropy RESidual Sum and Outlier test; MR-RAPS, Mendelian Randomization Robust Adjusted Profile Score; OR, odds ratio; CGRP, Calcitonin gene-related peptide; VIP, vasoactive intestinal peptide; 5-HT, 5-hydroxytryptamine.

## Conclusion

Our findings elucidate the positive causal effect of GERD on migraine and underscores the mediating role of depression in increasing the risk of migraine due to GERD. Effective control of GERD, particularly interventions targeting depression, may aid in preventing the occurrence of migraine. Future research should delve deeper into the specific pathophysiological mechanisms through which GERD affects migraine risk, facilitating the development of more effective drug targets or disease management strategies.

## Introduction

Migraine is a chronic neurovascular headache characterized by pulsating pain typically on one side of the head and lasts from hours to days [1]. Nausea, vomiting, visual disturbances, photophobia, and phonophobia often accompany migraines. Physical activities exacerbate headaches [2, 3]. Over 1 billion people globally are estimated to suffer from migraines, making it the third most prevalent disorder globally [4]. Its onset typically occurs in adolescence, with females experiencing three times the prevalence in males, which peaks in the forties and further declines [4]. Despite its medical importance, our understanding of migraine's pathophysiology remains limited. Some migraine sufferers may show a genetic predisposition, and certain environmental factors, including hormonal changes, weather fluctuations, bright lights, loud noises, and specific odors, may trigger migraines [5–7]. Lifestyle factors, such as diet, sleep, exercise, and gut microbiota, may influence migraine occurrence [8–10]. Psychological and social factors may be involved in the central sensitization of chronic pain [11]. Therefore, identifying and controlling these risk factors is crucial to prevent migraine attacks and reduce their frequency, duration, and severity [12].

Gastroesophageal reflux disease (GERD) includes recurrent acid reflux symptoms or GERD-specific complications, affecting approximately 20% of adults in high-income countries [13], with an increasing incidence. The primary symptoms of GERD involve the reflux of gastric contents (including acid, food, and bile) into the esophagus [14]. Several factors affect GERD's occurrence, including genetics, diet, smoking, alcohol consumption, obesity, and esophageal anatomical abnormalities [15]. Prolonged gastric content exposure can lead to structural and functional damage to the esophagus, making GERD a risk factor for esophageal ulcers, strictures, Barrett's esophagus, and esophageal adenocarcinoma [15]. GERD may lead to extraesophageal symptoms, including pharyngitis, oral inflammation, dental caries, tracheitis, and cough, which imposes a substantial physical, psychological, and medical burden on patients [16]. GERD's effects extend to the gastrointestinal tract and adjacent systems and involve the central nervous system. Existing perspectives suggest the link of migraines to gastrointestinal symptoms through the brain-gut axis, the bidirectional communication connecting the central nervous system with the gut [17]. This communication is achieved through interactions between the nervous, endocrine, and immune systems [18]. Inflammatory mediators, neuropeptides, immune cells, and gut microbiota are crucial in the brain-gut axis [19–21]. GERD-induced esophageal inflammation can promote the release of inflammatory mediators, such as interleukins and tumor necrosis factor, which may affect the gut-brain axis' function [19]. Enteroendocrine cells in the gut can release neuropeptides, including serotonin, dopamine, and gastrin, which affect neural transmission and regulation in the brain through

the gut-brain axis [19]. Abnormal gut microbiota proliferation or microbial species composition changes can affect the gut-brain axis [20].

Apart from the mechanisms mentioned above supporting the potential causal association between GERD and migraines, the causal link between migraines and GERD has been scarcely explored by existing observational studies. In a retrospective study of clinical data [17] involving 22,444 patients with migraine and 289,785 controls, Kim et al. found a higher incidence of GERD among patients with migraine (OR = 1.55, 95%CI: 1.45–1.66, p<0.001). Another multi-center study including 195 children with primary headaches (migraine or tension-type headache) [22] revealed the association of a higher prevalence of migraine without aura with patients with GERD (OR = 2.3, 95%CI: 1.2–4.4, p = 0.01). Similarly, Hormati et al. [23], in a cross-sectional study involving 109 patients with GERD and 232 controls, found a higher risk of migraines among patients with GERD (p < 0.001). However, these observational studies failed to establish the causal relationship and direction between GERD and migraines owing to a lack of validation from effective randomized controlled studies. Unraveling the causal relationship between migraines and GERD from observational studies is challenging and hindered by various confounding factors, often leading to inconclusive findings.

We adopted Mendelian randomization (MR), an epidemiological approach, to delve deeper into the true causal relationship between GERD and migraines. MR utilizes genetic variations as instrumental variables (IVs), aiding in a deeper understanding of the causal relationship between exposure to risk factors and disease outcomes [24, 25]. Genetic variations are randomly allocated during meiosis and remain stable throughout life; thus, MR is considered a natural randomized controlled trial (RCT) [26]. MR's unique characteristics make it less susceptible to reverse causation and confounding factors [26]. Initially, we conducted bidirectional two-sample MR (TSMR) to determine the causal relationship and direction between migraines and GERD. Subsequently, using multivariable MR (MVMR), we adjusted potential confounding factors, including smoking and alcohol consumption. In previous studies, it has been shown that GERD increases the risk of depression [27], and depression increases the risk of migraines [28]. However, no one has yet provided evidence to construct a complete disease progression curve for these three conditions. Therefore, we conducted mediation MR (MMR) analysis to validate if depression mediated the relationship between GERD and migraines by adjusting depression as a confounding factor in MVMR.

## Methods

### Study design

We utilized summarized data on GERD from a large-scale genome-wide association study (GWAS) and two large-scale GWAS on migraines. Fig 1 provides a detailed description of our study design. Our study was based on the fundamental assumptions of MR [29], that is, I) instrumental variables (IVs) are closely associated with the exposure factor; II) IVs are unrelated to any potential confounding factors, and III) the outcome can be only explained based on the association between the exposure factor and IVs. We adhered to reporting principles according to the STROBE-MR guidelines [30].

### Data sources

Summarized data on GERD were derived from a meta-analysis by Ong JS et al. [31], collected from the UK Biobank (UKBB) and the QSKIN study. Discovery analysis data for migraines was obtained from FinnGen [32]. Validation analysis data on migraines were sourced from the multi-ethnic GWAS meta-analysis published by Choquet et al. [33]. For MVMR analysis, including smoking and alcohol consumption, we utilized the data reported by the GWAS &

## (i) Basic assumptions of MR

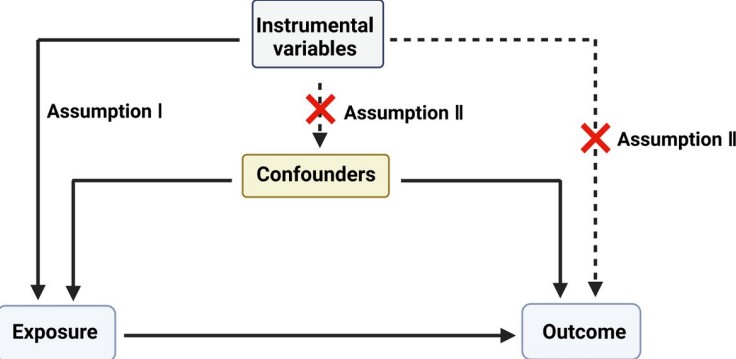

## (ii) MR analysis steps

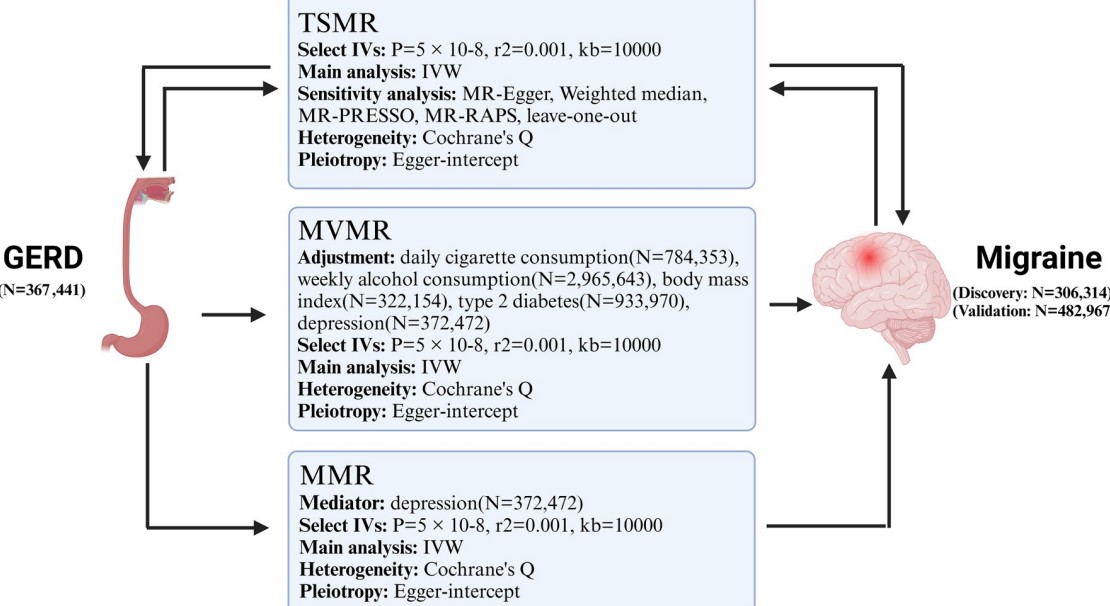

**Fig 1. A detailed description of study design.** MR, Mendelian randomization; GERD, gastroesophageal reflux disease; TSMR, Two-sample Mendelian randomization; MVMR, multivariable Mendelian randomization; MMR, mediation Mendelian randomization; IVW, inverse variance weighting; MR-Egger, MR-Egger regression; MR-PRESSO, MR Pleiotropy RESidual Sum and Outlier test; MR-RAPS, MR robust adjusted profile score.

Sequencing Consortium of Alcohol and Nicotine Use (GSCAN) [34]. The smoking phenotype comprised information on the daily cigarette consumption (CigDay) for current and former regular smokers, while the alcohol consumption phenotype utilized weekly alcohol consumption (DrnkWk) information. In MVMR analysis, summarized data for body mass index (BMI) were obtained from the Genetic Investigation of Anthropometric Traits (GIANT) [35], and those for Type 2 diabetes (T2D) were obtained from the Diabetes Genetics Replication and Meta-analysis (DIAGRAM) [36]. Summary data for depression used in MMR analysis were sourced from FinnGen [32]. Phenotypic details are described in Table 1, and more details can be found in the original studies. We solely utilized summary data for analysis; no additional ethics approval was required

**Table 1. Phenotypic details of this study.**

| Phenotype | Consortium | Population | Case | Control | Total sample size | Phenotype definition of case |
|---|---|---|---|---|---|---|
| GERD | UKBB | European | 78,707 | 354,285 | 367,441 | ICD-10 (FID: 41202/41204), ICD-9 (FID: 41203/41205), operative procedures (FID: 41200/41210), Use of GERD-related medications, Self-report (FID: 20002) |
| | QSkin | | | 13,156 | | self-reported heartburn and took one or more types of GERD-related medications |
| Migraine (discovery) | FinnGen | European | 18,477 | 287,837 | 306,314 | ICD-10 (G43), ICD-9 (346), ICD-8 (346) |
| Migraine (validation) | UKBB | European | 17,532 | 465,435 | 482,967 | Self-reported, ICD10 (G43) |
| | GERA | Mixed | 11,320 | 60,282 | 71,602 | ICD-9 (346.XX), ICD-10 (G43.XX) |
| CigDay | GSCAN | European | —— | —— | 784,353 | Daily cigarette consumption for current or ever smokers |
| DrnkWk | GSCAN | European | —— | —— | 2,965,643 | Weekly alcohol consumption |
| BMI | GIANT | European | —— | —— | 322,154 | $kg/m^2$ |
| T2D | DIAGRAM | European | 80,154 | 853,816 | 933,970 | Diagnosis according to WHO 1999 recommendations |
| Depression | FinnGen | European | 43,280 | 329,192 | 372,472 | ICD-10 (F3[2–3]|F341), ICD-9 (2961|2968|3004) |

GERD, gastroesophageal reflux disease; CigDay, daily cigarette consumption; DrnkWk, weekly alcohol consumption; BMI, body mass index; T2D, type 2 diabetes; UKBB, Unite Kingdom Biobank; QSkin, QSkin Study; GERA, Genetic Epidemiology Research in Adult Health and Aging; GSCAN, GWAS & Sequencing Consortium of Alcohol and Nicotine use; GIANT, Genetic Investigation of Anthropometric Traits; DIAGRAM, Diabetes Genetics Replication and Meta-analysis; FID, UKBB field ID; ICD International Classification of Diseases.

### Selection of genetic instruments

We rigorously used genome-wide thresholds (p<5.0E-8) [37] to screen single nucleotide polymorphisms (SNPs) strongly associated with GERD, CigDay, DrnkWk, BMI, T2D, and depression as IVs. When screening for IVs in the discovery set of migraines, the number of SNPs obtained using the threshold of p<5.0E-8 was too low, so we relaxed the threshold to p<5.0E-7. The selection threshold for IVs in the validation set of migraines was p<5.0E-8. We removed linkage disequilibrium between genetic loci using a clumping window of 10000 kb and $r^2$ of 0.001 to avoid violating assumptions II and III of MR [29]. Considering the significant missing IVs for migraines in GERD, $r^2>0.9$ was used to search for proxy SNPs [38], and we found one proxy SNP each for the discovery and validation sets of migraines. We calculated the F-statistic for each SNP and the total F-statistic for all SNPs to assess the strength of association between SNPs and the corresponding phenotypes [39]. SNPs with an F-statistic less than 10 were removed, and those with an F-statistic greater than 10 indicate a sufficiently strong association [39]. The formula for calculating the F-statistic is provided in Supplementary Methods. We searched IVs related to GERD, migraines, and depression in PhenoScanner [40] (S1-S4 Tables in S1 File). SNPs directly related to the results of MR analysis or potentially confounding factors were excluded. We harmonized the information on SNPs for exposure and outcome and removed palindromic SNPs that could not be identified in the correct direction [25]. Steiger tests were conducted for the coordinated data to ensure the correctness of the causal direction, and SNPs with a greater effect on the outcome than the exposure were removed [41]. The detailed steps for selecting IVs in this study are described in S1 Fig.

### Two-sample mendelian randomization

The inverse variance weighted (IVW) random effects model is the primary method for TSMR analysis. IVW considers heterogeneity and correlation of genetic variants [42]. We employed MR-Egger regression [43] and weighted median [44] for primary sensitivity analyses. MR-Egger regression allows all genetic variants to violate the IV assumptions but may yield false-

negative results [43]. The weighted median provides reliable causal estimates when SNPs greater than 50% are valid IVs [44]. We used the MR Robust Adjusted Profile Score (MR-RAPS) [45] and MR Pleiotropy RESidual Sum and Outlier test (MR-PRESSO) [46] for sensitivity analyses. MR-RAPS considers horizontal pleiotropy and weak instrument bias [45]. MR-PRESSO detects potential horizontal pleiotropy from distortion tests and provides causal estimates based on IVW after identified outliers are removed [46]. In TSMR, MR-PRESSO estimates were used after removing outliers. $I^2$ and Cochran's Q value to assess heterogeneity [47, 48], where $I^2 > 90\%$ indicates considerable reliability of the results [47]. The magnitude of heterogeneity was evaluated using the intercept from MR-Egger regression [43]. Leave-one-out analysis was performed to determine if individual SNPs affected causal estimates [49]. Finally, the mRnd tool [50] was used to calculate the statistical power of the TSMR analysis.

## Multivariable mendelian randomization

The primary analytical approach for MVMR is the random effects model of IVW [51]. We employed Cochran's Q and the intercept from MR-Egger regression, respectively, to assess heterogeneity and pleiotropy [43]. The "MVMR" package was used to compute conditional F-statistics for MVMR to evaluate the strength of IVs [52].

## Mediation mendelian randomization

MMR analysis was conducted using a two-step approach. Its detailed design is outlined in S2 Fig. We computed the following causal effects: 1) the overall effect of GERD on migraines ($\beta0$); 2) the direct effect of GERD on depression ($\beta1$); 3) the direct effect of depression on migraines ($\beta2$). The mediation effect ($\beta3$) was calculated using the following formula: $\beta3 = \beta1 \times \beta2$ [53], and the percentage of mediation effect was calculated as ($\beta3 / \beta0$)*100%. IVW was the primary method used to compute causal effects, while MR-Egger, weighted median, and leave-one-out were performed to analyze the sensitivity [54]. Scatter plots show the results from different causal estimation models [55]. Heterogeneity analysis was performed using Cochran's Q value. MR-Egger regression intercept and funnel plot were used to assess pleiotropy's magnitude [43].

## Software support

All data were analyzed using the R software (version 4.2.3). For TSMR analysis, the "TwoSampleMR" package [55] and the "MR-RAPS" package [45] were used. The "TwoSampleMR" package [55] and the "MVMR" package [52] were used for MVMR analysis. We utilized the "TwoSampleMR" package for MMR analysis [55].

# Results

## Genetic instruments

In the TSMR analysis, 64 and 63 SNPs were identified as IVs for MR analysis from GERD to migraines (discovery and validation sets, respectively), and 6 and 12 SNPs were considered as IVs for the MR analysis from migraines to GERD (discovery and validation sets, respectively). In the MVMR analysis, after adjusting for CigDay, 55 and 54 SNPs were retained as IVs for GERD in the discovery and validation sets of migraines, respectively. After adjusting for DrnkWk, 55 and 54 SNPs were retained as IVs for GERD in the discovery and validation sets of migraines, respectively. After adjusting for BMI, 55 and 54 SNPs were retained as IVs for GERD in the discovery and validation sets of migraines, respectively. After adjusting for T2D, 42 and 40 SNPs were retained as IVs for GERD in the discovery and validation sets of migraines,

respectively. After adjusting for depression, 62 and 60 SNPs were retained as IVs for GERD in the discovery and validation sets of migraines, respectively. In the MMR analysis, 52 SNPs were identified as IVs for GERD to depression. We identified 14 and 12 SNPs as IVs for depression to migraines (discovery and validation sets). The selected IVs explained variance ranging from 2.35% to 2.87% for GERD, 1.08% (discovery set) and 2.89% (validation set) for migraines, 1.18% (discovery set), and 1.32% (validation set) for depression. In TSMR and MMR analyses, no SNP had an F-value less than 10 (S5 Table in S1 File), and the total F values were greater than 10 (S6 Table in S1 File). In the MVMR analysis, conditional F-statistics were all greater than 10 (S7 Table in S1 File), indicating that the selected IVs had strong statistical power.

## Results from the TSMR analysis

The MR calculation results are mainly presented in the form of odds ratios (OR). A bidirectional TSMR analysis was conducted, examining the relationship between migraines and GERD (Figs 2 and 3). Genetically predicted GERD increased the risk of developing migraines (IVW: discovery, OR = 1.49, 95%CI: 1.34–1.66, p = 3.70E-13; validation, OR = 1.28, 95%CI: 1.17–1.40, p = 7.41E-08). Causal estimates from other methods were consistent with IVW. However, genetically predicted migraines did not increase the risk of GERD (IVW: discovery, OR = 1.07, 95%CI: 0.98–1.17, p = 0.1139; validation, OR = 0.99, 95%CI: 0.95–1.04, p = 0.7676).

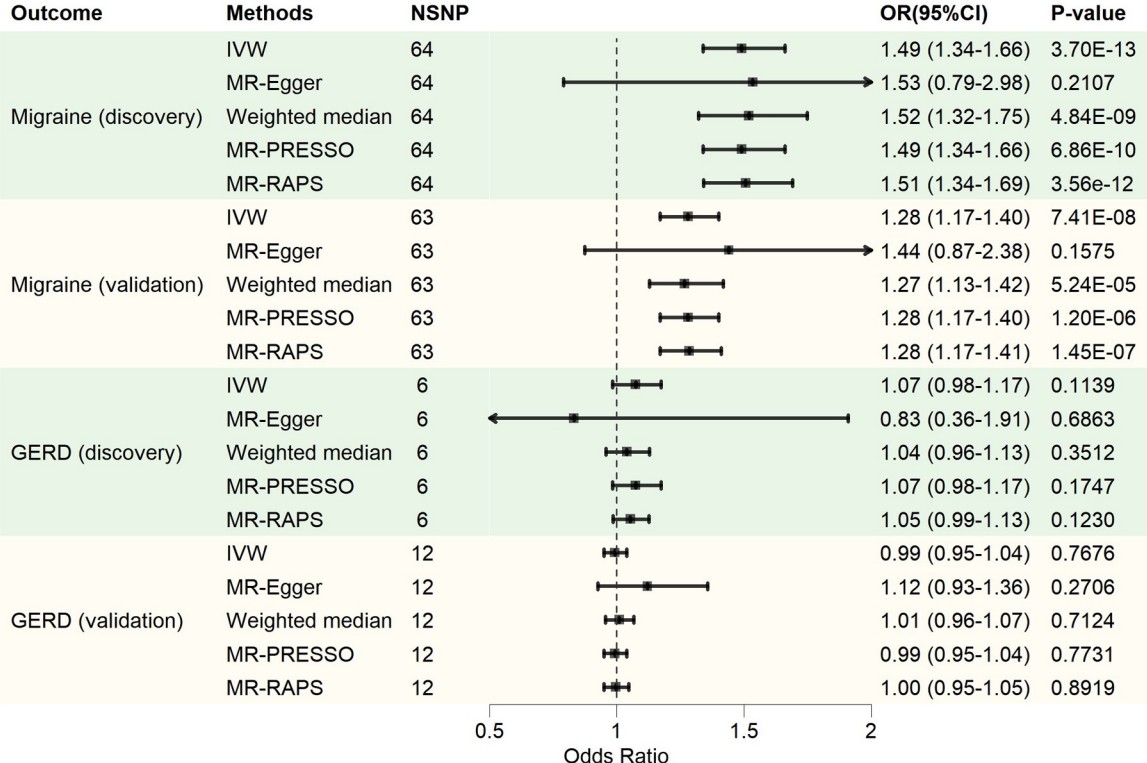

**Fig 2. Two-sample Mendelian randomization analysis causal estimation.** NSNP, number of SNPs; OR, odds ratio; 95% CI, 95% confidence interval; GERD, gastroesophageal reflux disease; IVW, inverse variance weighting; MR-Egger, MR-Egger regression; MR-PRESSO, MR Pleiotropy RESidual Sum and Outlier test; MR-RAPS, MR robust adjusted profile score.

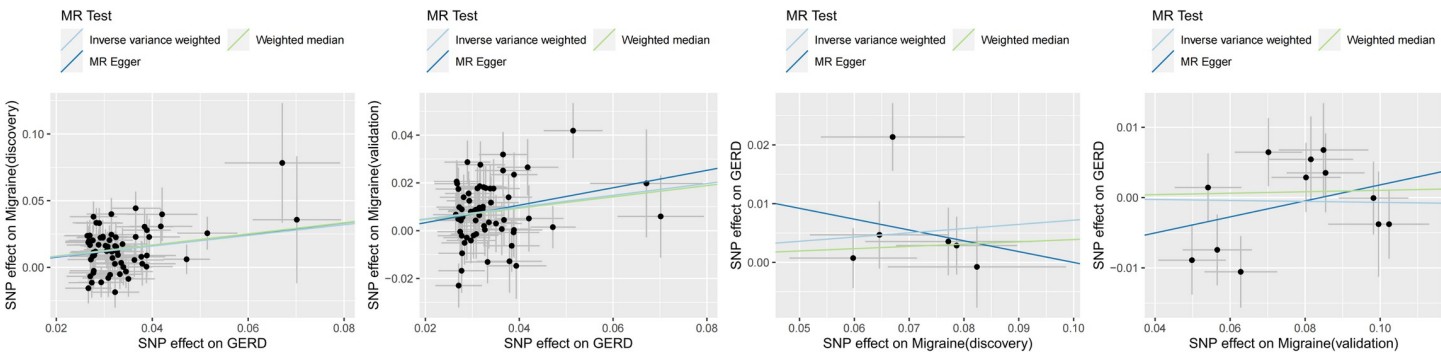

**Fig 3. Scatterplots for the results of the two-sample Mendelian randomization analysis.** GERD, gastroesophageal reflux disease; MR, Mendelian randomization; SNP, single nucleotide polymorphism.

## Results from the MVMR analysis

In the MVMR analysis, we adjusted for CigDay, DrnkWk, BMI, T2D, and depression. The positive causal effect of GERD on migraines remained Even after adjusting for these confounding factors (Fig 4), suggesting that the effect of GERD on migraines is independent of these confounding factors.

Table 2 presents the results of heterogeneity and pleiotropy analyses for TSMR and MVMR analyses. Heterogeneity was observed in MR analyses; however, in all analyses, the MR-Egger intercept did not detect horizontal pleiotropy. No significant horizontal pleiotropy or individual SNP with a significant effect on the results was observed in the funnel plots and leave-one-out results of the TSMR analysis (S3 and S4 Figs). The results of TSMR and MVMR analyses demonstrated strong robustness, indicating minimal susceptibility to horizontal pleiotropy.

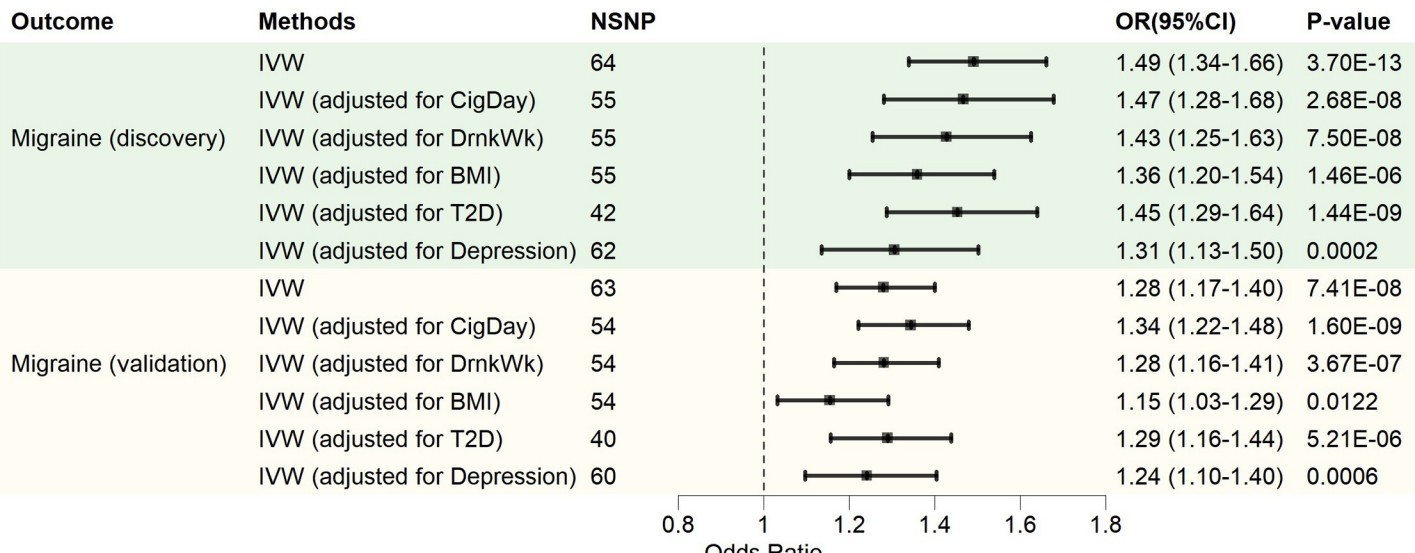

| Outcome | Methods | NSNP | | OR(95%CI) | P-value |
|---|---|---|---|---|---|
| Migraine (discovery) | IVW | 64 | | 1.49 (1.34-1.66) | 3.70E-13 |
| | IVW (adjusted for CigDay) | 55 | | 1.47 (1.28-1.68) | 2.68E-08 |
| | IVW (adjusted for DrnkWk) | 55 | | 1.43 (1.25-1.63) | 7.50E-08 |
| | IVW (adjusted for BMI) | 55 | | 1.36 (1.20-1.54) | 1.46E-06 |
| | IVW (adjusted for T2D) | 42 | | 1.45 (1.29-1.64) | 1.44E-09 |
| | IVW (adjusted for Depression) | 62 | | 1.31 (1.13-1.50) | 0.0002 |
| Migraine (validation) | IVW | 63 | | 1.28 (1.17-1.40) | 7.41E-08 |
| | IVW (adjusted for CigDay) | 54 | | 1.34 (1.22-1.48) | 1.60E-09 |
| | IVW (adjusted for DrnkWk) | 54 | | 1.28 (1.16-1.41) | 3.67E-07 |
| | IVW (adjusted for BMI) | 54 | | 1.15 (1.03-1.29) | 0.0122 |
| | IVW (adjusted for T2D) | 40 | | 1.29 (1.16-1.44) | 5.21E-06 |
| | IVW (adjusted for Depression) | 60 | | 1.24 (1.10-1.40) | 0.0006 |

**Fig 4. MVMR analysis causal estimation.** NSNP, number of SNPs; OR, odds ratio; 95% CI, 95% confidence interval; IVW, inverse variance weighting; CigDay, cigarettes per day; DrnkWk, alcoholic drinks per week; BMI, body mass index; T2D, type 2 diabetes.

## Results from the MMR Analysis

Depression mediated the positive causal effect of GERD on migraine (Table 3). In the discovery stage, the total effect (β0) of GERD on migraine was 0.399; the direct effect (β1) of GERD on depression was 0.348, and the direct effect (β2) of depression on migraine was 0.329. Depression's mediating effect accounted for 28.72% (95% CI: 17.82–34.98) of the total effect of GERD on migraine. Similar mediation was observed in the validation analysis, with the mediating effect of depression accounting for 26.38% (95% CI: 9.01–34.48) of the total effect of GERD on migraine. Across all steps of the MMR analysis, the results of various supplementary methods remained consistent with the IVW findings (S5 Fig). Heterogeneity was observed in some MR results (S8 Table in S1 File). However, neither the MR-Egger intercept (S8 Table in S1 File) nor the funnel plot (S6 Fig) showed significant horizontal pleiotropy. The leave-one-out method did not reveal significant outliers (S7 Fig). These findings indicated the robustness of the results of our MMR analysis.

In summary, our study presents the following findings: GERD increases the risk of migraine, while migraine does not increase the risk of GERD. Additionally, GERD increases the risk of depression, and depression further increases the risk of migraine. These analytical results are robust, supported by different computational models, and are replicable for further analysis.

## Discussion

We are the first to investigate the causal relationship between GERD and migraine using MR. We conducted TSMR, MVMR, and MMR analyses to explore the relationship between these two conditions by leveraging the largest-scale GWAS summary data for GERD and two extensive GWAS summary data for migraine. Our findings provide robust evidence that indicates GERD may increase the risk of migraine, but migraine does not increase the risk of GERD. Interestingly, depression may act as an intermediate factor in increasing the risk of migraine associated with GERD.

**Table 2. Heterogeneity and pleiotropy of MR analysis.**

| Exposure | Outcome | Cochrane's Q | p-value | Egger-intercept | p-value | I²(%) |
|---|---|---|---|---|---|---|
| Univariate MR | | | | | | |
| GERD | Migraine (discovery) | 87.35 | 0.0229 | -0.0009 | 0.9314 | 97.42 |
| GERD | Migraine (validation) | 91.55 | 0.0087 | -0.0039 | 0.6389 | 97.42 |
| Migraine (discovery) | GERD | 10.14 | 0.0714 | 0.0184 | 0.5760 | 97.18 |
| Migraine (validation) | GERD | 15.04 | 0.1809 | -0.0096 | 0.2326 | 98.52 |
| Multivariate MR | | | | | | |
| GERD (adjustment of CigDay) | Migraine (discovery) | 98.68 | 0.0021 | -0.0005 | 0.9433 | — |
| GERD (adjustment of DrnkWk) | | 128.69 | 1.75E-05 | 0.0002 | 0.9412 | |
| GERD (adjustment of BMI) | | 207.86 | 9.29E-09 | 0.0011 | 0.6830 | |
| GERD (adjustment of T2D) | | 150.76 | 0.0072 | 0.0004 | 0.8878 | |
| GERD (adjustment of Depression) | | 91.20 | 0.0144 | -0.0007 | 0.9317 | |
| GERD (adjustment of CigDay) | Migraine (validation) | 75.98 | 0.0937 | -0.0025 | 0.6231 | |
| GERD (adjustment of DrnkWk) | | 101.39 | 0.0054 | 0.0016 | 0.4123 | |
| GERD (adjustment of BMI) | | 260.25 | 5.57E-15 | 0.0027 | 0.2754 | |
| GERD (adjustment of T2D) | | 181.73 | 2.63E-05 | 0.0017 | 0.4661 | |
| GERD (adjustment of Depression) | | 101.75 | 0.0011 | 4.40E-05 | 0.9949 | |

I², I squared; GERD,gastroesophageal reflux disease; CigDay, daily cigarette consumption; DrnkWk, weekly alcohol consumption; BMI, body mass index; T2D, type 2 diabetes.

**Table 3. Mediation effect of depression on the GERD to migraine.**

| Exposure | Mediator | Outcome | β0 | | | β1 | | | β2 | | | β3 | | | |
|---|---|---|---|---|---|---|---|---|---|---|---|---|---|---|---|
| | | | Beta | SE | P-value | Beta | SE | P-value | Beta | SE | P-value | Beta | SE | Prop % (95%CI) | P-value |
| GERD | Depression | Migraine (discovery) | 0.399 | 0.055 | 3.70E-13 | 0.348 | 0.041 | 2.37E-17 | 0.329 | 0.084 | 9.52E-05 | 0.115 | 0.032 | 28.72(17.82–34.98) | 0.0004 |
| | | Migraine (validation) | 0.247 | 0.046 | 7.41E-08 | | | | 0.159 | 0.060 | 0.0075 | 0.065 | 0.026 | 26.38(9.01–34.48) | 0.0111 |

β0, total effect of GERD to migraine; β1, direct effects of GERD to depression; β2, direct effects of depression to migraine; β3, mediation effect of GERD to migraine; Beta, beta values for causal estimates; SE, standard errors of causal estimates; P-value, P-value of Beta; Prop, proportion; 95%CI, 95% confidence interval.

GERD and migraine are two highly prevalent diseases that impose a physiological, psychological, and medical burden on patients. Limited evidence exists on the association of GERD with migraine but several preliminary observational studies compelled us to reconsider their relationship. For instance, observational studies by Lenglart et al. [22] and Hormati [23] indicated a significant increase in the prevalence of migraine among patients with GERD, consistent with our findings. However, the results of studies of Kim et al. [17] and Katić et al. [56] suggest an elevated prevalence of GERD among patients with migraine, contradicting the results of our MR analysis. The advantages of MR studies support our reasonable skepticism regarding the observed phenomenon of increased GERD incidence associated with migraines in observational studies. First, MR studies allow for drawing causal inferences at the genetic level [24, 25], which observational studies cannot achieve. Second, rigorously designed MR studies are less susceptible to reverse causation and confounding factors [26], which are challenging to avoid in observational studies. Finally, our MR study comprised a large sample size, and all results in the validation cohort were replicated, enhancing the robustness of the statistical power of the findings.

Migraine's etiology is extremely complex, involving multiple factors, including the "gut-brain axis" [57]. Caution is warranted to explain how GERD increases the risk of migraine. Migraine is considered a neurovascular disorder originating from the central nervous system, involving the activation and sensitization of the trigeminovascular system [1]. This process involves the release of pro-inflammatory neuropeptides and neurotransmitters, ultimately leading to vascular dilation, mast cell degranulation, plasma extravasation, and tissue edema. When gastric contents reflux into the esophagus of patients with GERD, chronic esophageal stress and inflammation may occur [58], triggering a state of "aseptic inflammation" in the intracranial meninges which leads to sensitization and activation of trigeminal meningeal nociceptors [59, 60], precipitating into migraine. Autonomic nervous system dysfunction is an important pathophysiological basis for migraine, involving the activation of the hypothalamic and central autonomic networks during the prodromal phase of migraine and enhanced connectivity between these regions during interictal periods [61]. Inflammation associated with GERD or esophageal dysmotility may stimulate the vagus or sympathetic nervous system, resulting in autonomic dysregulation, particularly an imbalance between the functions of the parasympathetic and sympathetic nervous systems [62]. This imbalance may result in vascular dilation, altered blood flow, and abnormal neuronal activity, further enhancing the sensitization of pain centers and increasing the risk of migraine [63].

Some neuropeptides and neurotransmitters are implicated in promoting the risk of migraine occurrence in GERD, including calcitonin gene-related peptide (CGRP), vasoactive intestinal peptide (VIP), and serotonin. CGRP plays various roles in neurogenic inflammation, with evidence indicating its ability to promote plasma extravasation, mast cell degranulation, and release of pro-inflammatory substances [64]. Elevated CGRP levels have been observed in

patients with GERD [65], suggesting that GERD may promote migraine through this pathway. VIP, as a potent vasodilator, regulates mast cell degranulation and the production of pro-inflammatory cytokines, such as interleukin-6 [66]. VIP levels increase in patients with GERD [67] and may promote vasodilation and mast cell degranulation in the dura mater, leading to migraine. Serotonin, or 5-hydroxytryptamine (5-HT), is an important neurotransmitter and vasoactive substance synthesized primarily by enterocytes and neurons in the gut [68]. It has been implicated in neurological and psychological disorders, such as depression, anxiety, migraine, and schizophrenia [69]. Gastric and intestinal mucosal epithelial damage caused by GERD may reduce 5-HT levels in the body [70] and promote migraine.

GERD may lead to dysbiosis of the gut microbiota [71], thereby affecting the levels of tumor necrosis factor-alpha and serotonin in the trigeminal nociceptive system, involved in the normal mechanical nociception and pathogenesis of migraine [72, 73]. Many studies suggest an association between GERD and psychiatric symptoms, particularly depression, with up to one-third of patients with GERD reportedly experiencing depression [74, 75]. Existing evidence hints at depression as a risk factor for migraine [76, 77]. Our mediation MR analysis successfully connects GERD, depression, and migraine, forming a complete pathway of disease development. Depression crucially promotes migraine occurrence in GERD, with serotonin may playing a key role [69]. Recent evidence suggests that immune cells may serve as a potential tertiary structure between migraine and the gut-brain axis. In migraine, the number of regulatory T cells and cytotoxic T cells decreases while helper T cells increase. Peripheral regulatory T cells levels are considered potential biomarkers for the treatment of migraine [19, 78]. GERD may induce changes in the mucosal immune system and promote migraine occurrence [79]. Finally, shared genetic factors may underlie predisposition to migraine and GERD. However, caution should be exercised in interpreting these results owing to a lack of support from RCTs. Future research should explore the exact underlying pathophysiological mechanisms and potential interventions.

In summary, patients with GERD showing genetic susceptibility may possess certain traits involving immunity, inflammation, neurotransmitters, and psychological factors, which leads to a higher prevalence of migraine. Our findings support the need for routine assessment and early identification of migraines in patients with GERD in clinical practice. Existing observational studies have emphasized the potential benefits of this approach. For instance, a case-control study by Naoum et al. [80] found a higher occurrence of headaches in patients with gastrointestinal issues, with improvement in headaches within one month after treating gastrointestinal symptoms (p < 0.001). Patients with GERD should prioritize emotional management, especially avoiding depression, which may help reduce the incidence of migraines and ultimately improve their quality of life.

Some limitations of our study need to be acknowledged. First, we cannot entirely eliminate the risk of pleiotropic bias inherent to MR studies (26). Second, most of the samples we used were of European descent, limiting the generalizability of our findings. Third, a risk of sample overlap between the dataset for depression and the discovery set for migraine existed, although there was no overlap with the validation set for migraine; our final conclusions were unaffected. Lastly, despite the sex-based differences in prevalence and different subtypes of migraine, owing to the limitations in existing data, we could not conduct a sex-stratified analysis and an analysis based on migraine subtypes.

Despite the limitations, our findings indicated that GERD increased the risk of migraine but migraine did not increase the risk of GERD. GERD enhanced the risk of depression, and depression further enhanced the risk of migraine. These analytical results were robust, supported by different computational models, and replicable for further analysis.

## Conclusion

In conclusion, our MR study provides compelling evidence confirming the effect of genetic susceptibility to GERD on increasing the risk of migraine. In contrast, genetic susceptibility to migraine did not alter the risk of GERD. The viewpoint from observational studies regarding migraine potentially increasing the risk of GERD may be misconstrued. Our study emphasizes the mediating role of depression in this causal association, suggesting that effective control of GERD in patients, particularly in managing depressive symptoms, may help prevent the occurrence of migraines. Future research should focus on clinical and basic RCTs to explore the pathophysiological mechanisms by which GERD alters migraine risk and strive to develop more effective drug targets or disease management strategies based on these findings.

## Supporting information

**S1 Fig.**
(TIF)

**S2 Fig.**
(TIF)

**S3 Fig.**
(TIF)

**S4 Fig.**
(TIF)

**S5 Fig.**
(TIF)

**S6 Fig.**
(TIF)

**S7 Fig.**
(TIF)

**S1 File. Supplementary tables.**
(XLSX)

**S1 Graphical abstract. Abbreviations: GERD, gastroesophageal reflux disease; IV, instrumental variable; TSMR, two-sample Mendelian randomization; MVMR, multivariate Mendelian randomization; MMR, mediation Mendelian randomization.**
(TIF)

## Acknowledgments

We are grateful to the researchers and organizations who shared genome-wide association study summary data involved in this study.

## Author Contributions

**Conceptualization:** Zixiong Shen, Yewen Bian, Yao Huang, Hao Chen, Liuying Li.

**Data curation:** Zixiong Shen, Yewen Bian, Wenhua Zhou, Hao Chen, Xia Zhou.

**Formal analysis:** Zixiong Shen, Yao Huang, Hao Chen.

**Funding acquisition:** Liuying Li.

**Resources:** Yewen Bian.

**Software:** Zixiong Shen.

**Supervision:** Zixiong Shen, Wenhua Zhou, Hao Chen, Xia Zhou, Liuying Li.

**Validation:** Zixiong Shen, Yewen Bian, Yao Huang, Wenhua Zhou, Xia Zhou, Liuying Li.

**Visualization:** Zixiong Shen, Xia Zhou.

**Writing – original draft:** Zixiong Shen, Yao Huang, Hao Chen.

**Writing – review & editing:** Zixiong Shen, Yewen Bian, Yao Huang, Wenhua Zhou, Xia Zhou, Liuying Li.

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
