## [Decision Letter · Decision Letter 0]

10 Jan 2024

PONE-D-23-39298Gastroesophageal reflux disease increases migraine risk with depression as an important mediator: A Mendelian randomization studyPLOS ONE

Dear Dr. Shen,

Thank you for submitting your manuscript to PLOS ONE. After careful consideration, we feel that it has merit but does not fully meet PLOS ONE’s publication criteria as it currently stands. Therefore, we invite you to submit a revised version of the manuscript that addresses the points raised during the review process.

We look forward to receiving your revised manuscript.

Kind regards,

Y-h. Taguchi, Dr. Sci.

Academic Editor

PLOS ONE

Journal Requirements:

Whilst you may use any professional scientific editing service of your choice, PLOS has partnered with both American Journal Experts (AJE) and Editage to provide discounted services to PLOS authors. Both organizations have experience helping authors meet PLOS guidelines and can provide language editing, translation, manuscript formatting, and figure formatting to ensure your manuscript meets our submission guidelines. To take advantage of our partnership with AJE, visit the AJE website (http://aje.com/go/plos) for a 15% discount off AJE services. To take advantage of our partnership with Editage, visit the Editage website (www.editage.com) and enter referral code PLOSEDIT for a 15% discount off Editage services. If the PLOS editorial team finds any language issues in text that either AJE or Editage has edited, the service provider will re-edit the text for free.

This study was supported by The Key Research and Development Plan (Major Science and Technology Special Project) of the Sichuan Provincial Science and Technology Department (No.2022YFS0392) and the Key Discipline Construction Project of the Sichuan Provincial Administration of Traditional Chinese Medicine (No.202072).

Additional Editor Comments:

Please revise the manuscript with addressing all the concerns raised by the reviewers.

Reviewers' comments:

Reviewer's Responses to Questions

**Comments to the Author**

1. Is the manuscript technically sound, and do the data support the conclusions?

Reviewer #1: Yes

Reviewer #2: Yes

Reviewer #3: Yes

2. Has the statistical analysis been performed appropriately and rigorously? 

Reviewer #1: Yes

Reviewer #2: Yes

Reviewer #3: I Don't Know

3. Have the authors made all data underlying the findings in their manuscript fully available?

Reviewer #1: Yes

Reviewer #2: Yes

Reviewer #3: Yes

4. Is the manuscript presented in an intelligible fashion and written in standard English?

Reviewer #1: Yes

Reviewer #2: Yes

Reviewer #3: Yes

5. Review Comments to the Author

Reviewer #1: Thank you for inviting me to review this intriguing manuscript. Zixiong Shen et al.'s study provides new evidence from a Mendelian randomization perspective supporting the increased risk of migraines due to gastroesophageal reflux disease (GERD). The study's design and methodology are well-executed, the interpretations of the results are appropriate, and the manuscript's English writing is of good quality. They also conducted a mediation Mendelian randomization analysis, providing evidence suggesting the role of depression in the path from GERD to migraines. This is an interesting finding that reveals a potential mechanism by which gastrointestinal diseases impact brain disorders.

I particularly appreciate Figure 1, Figure S1 and Figure S2, where the authors clearly illustrate their research design. Overall, this is an excellent manuscript, but I have two minor suggestions:

In Table 1, the definitions of various phenotypes used in the GWAS data should have a more detailed range based on the International Classification of Diseases.

If possible, I suggest the authors include leave-one-out results, funnel plots, and scatter plots for the mediation Mendelian randomization analysis in the supplementary methods section.

Reviewer #2: 24 December 2023

The review on the manuscript, titled ‘Gastroesophageal reflux disease increases migraine risk with depression as an important mediator: A Mendelian randomization study’ by Shen Z et al., submitted to Plos One

Manuscript ID: PONE-D-23-39298

Dear Authors,

Migraine and gastroesophageal reflux disease (GERD) are both common disorders that affect the quality of life of many people. However, the relationship between them is unclear and controversial. The possible mechanisms and directions of causality are also unknown. Observational studies are insufficient to establish a causal relationship between migraine and GERD because they may be confounded by environmental and lifestyle factors. Moreover, the role of depression, which is often comorbid with both migraine and GERD, is also unclear. Therefore, a more rigorous and robust method is needed to explore the causal link and the potential mediators between these two disorders. In the present research article titled ‘Gastroesophageal reflux disease increases migraine risk with depression as an important mediator: A Mendelian randomization study,’ Shen and colleagues used the mendelian randomization (MR) method to assess the causal relationship between migraine and GERD

The strength of this study lies in that it used a robust method called MR to investigate the causal relationship between GERD and migraine, with depression as a mediator. The study employed a large sample size and multiple sensitivity analyses to ensure the validity of the results. The authors also used various statistical methods to assess heterogeneity and pleiotropy, which are potential sources of bias in MR studies. The study provides valuable insights into the potential causal relationship between GERD and migraine and the role of depression in this association.

In general, I think the idea of this article is really interesting, and the authors’ fascinating observations on this topic may be of interest to the readers of Plos One. However, some comments, as well as crucial evidence that should be included to support the authors’ argumentation, need to be addressed to improve the quality of the manuscript, its adequacy, and its readability prior to its publication in the present form. My overall opinion is to publish this research article after the authors have carefully considered the comments and suggestions.

Please consider the following comments.

1. I recommend revising the title. Please present a title that conveys the most important message of this manuscript. Suggestions: "Unraveling the Causal Relationship Between Migraine and Gastroesophageal Reflux Disease: A Mendelian Randomization Study"; "Migraine and GERD: Disentangling the Complex Connection with Depression as a Mediator" [1 –3].

2. A graphical abstract that visually summarizes the main findings of this manuscript is highly recommended.

3. Abstract: I would like the authors to make as much effort in this section for the remainder of the manuscript. Please expand the abstract in 200 words (preferably 200–220 words; max. 250), although the journal allows 300 words [4], focusing on proportionally presenting the background, methods, results, and conclusion (without the headings of the subsections). The background should include the general background (one to two sentences), the specific background (two to three sentences), and "the current issue addressed to this study" (one sentence), leading to the objectives. Additionally, the methods should clarify the authors’ approach, such as the study design and variables, to solve the problem and/or make progress on the problem. The results should be close to a single sentence, thus putting the results in a more general context. The conclusion should open with one sentence describing the main result using such words as “Here we show,” which should be followed by statements such as the potential and the advance this study has provided in the field, and finally a broader perspective (two to three sentences) readily comprehensible to a scientist in any discipline [5–8].

4. Keywords: Please list as many keywords as allowed by the journal, chosen from Medical Subject Headings (MeSH) [9], and use as many keywords as possible in the title and in the first two sentences of the abstract [7,8].

5. Introduction: The authors need to fully expand this section with several paragraphs made up of about 1000 words, introducing information on the main constructs of this protocol, which should be understood by a reader in any discipline, and making it persuasive enough to put forward the main purpose of the current research the authors have conducted and the specific purpose the authors have intended by this study. I would like to encourage the authors to present the introduction starting with the general background, proceeding to the specific background, and finally, the current issue addressed to this study, leading to the objectives. These main structures should be organized in a logical and cohesive manner [10]. In this regard, the manuscript would greatly benefit from incorporating a discussion of the following topics. The following studies, but not limited to, may enhance the value of this manuscript [11–17].

6. Methods: I recommend opening this section with a short introductory paragraph and citing more references to ensure the reliability and integrity of the evidence in the study design that the authors built and the methodology they have decided to apply.

7. Results: I suggest presenting the independent results section and closing this section with a paragraph that puts the results into a more general context.

8. Discussion: I would like the authors to present an independent section without subheadings with several paragraphs of about 1500 words, beginning this section with an introduction and providing a summary of the previous section. Then, I expect the authors to develop arguments clarifying the potential of this study as an extension of the previous work, the implications of the findings, how this study could facilitate future research, the ultimate goal, the challenge, the knowledge and technology necessary to achieve this goal, the statement about this field in general, and the importance of this line of research. It is particularly important to present the limitations, merits, and potential translation of this study into clinical practice [18,19].

9. Conclusion: I believe that presenting this section with 150–200 words would benefit from a single paragraph that presents some thoughtful and in-depth considerations by the authors as experts to convey the main message. The authors should make an effort to explain the theoretical implications and translational applications of their research. To understand the significance of this field, I believe it is necessary to discuss theoretical and methodological avenues in need of refinement as well as future research directions.

10. References: Please cite more references. An article like this typically cites more than 60 to 70 references.

Overall, the manuscript contains four figures, two table, and 48 references. I believe that the merit of this manuscript lies in its contribution to the understanding of the causal relationship between GERD and migraine, with depression as a mediator. The study used a robust MR method and a large sample size to provide strong evidence supporting the notion that GERD increases the risk of migraine, while migraine does not have a causal effect on GERD. The study also highlights the pivotal role of depression as a bridge in this relationship, with depression mediating a significant proportion of the effect of GERD on migraine. The findings of this study have important implications for the treatment and management of patients with GERD and migraine, particularly those with comorbid depression. The study suggests that interventions targeting GERD may help reduce the risk of migraine and that addressing depression may be an important component of such interventions. Overall, this study provides valuable insights into the complex interplay between gastrointestinal diseases and brain disorders and may pave the way for personalized treatment approaches in the future. I hope that after careful revision, the manuscript meets the journal’s high standards for publication.

I declare no conflict of interest regarding this manuscript.

Best regards,

Reviewer

References:

1. https://plos.org/resource/how-to-write-a-great-title/

2. https://www.nature.com/nature-index/news-blog/how-to-write-a-good-research-science-academic-paper-title

3. https://www.indeed.com/career-advice/career-development/catchy-title

4. https://journals.plos.org/plosone/s/submission-guidelines

5. https://www.scribbr.com/dissertation/abstract/

6. https://writing.wisc.edu/handbook/assignments/writing-an-abstract-for-your-research-paper/

7. https://doi.org/10.5812/ijem.100159

8. https://doi.org/10.4103/sja.SJA_685_18

9. https://meshb.nlm.nih.gov/

10. https://dept.writing.wisc.edu/wac/writing-an-introduction-for-a-scientific-paper/

11. https://doi.org/10.3390/biomedicines11112979

12. https://doi.org/10.3390/cells12222649

13. https://doi.org/10.3390/biomedicines10010076

14. https://doi.org/10.17219/acem/166476

15. https://doi.org/10.1007/s12035-023-03623-1

16. https://doi.org/10.3389/fnins.2023.1233601

17. https://doi.org/10.3390/biomedicines9080897

18. https://doi.org/10.3163/1536-5050.103.2.001

19. https://www.scribbr.com/dissertation/discussion/

Reviewer #3: The authors of a study concluded that depression can significantly increase the likelihood of developing gastroesophageal reflux disease (GERD) and migraine. According to the manuscript, the sentence reads as follows: 'Depression mediated 28.72% (26.38% in the validation dataset) of the effect of GERD on migraine.'

I have a concern about the exclusion of other variables, such as smoking, alcohol consumption, and obesity, which could also have an impact on the development of GERD and migraine.

Furthermore, it might be preferred to exclude depression as a variable and focus only on the causal effect of GERD on migraine."

6. PLOS authors have the option to publish the peer review history of their article (what does this mean?). If published, this will include your full peer review and any attached files.

Reviewer #1: No

Reviewer #2: No

Reviewer #3: No

---

## [Author Response · Author response to Decision Letter 0]

10 Apr 2024

Response to academic editor

Dear Editor, thank you for giving us the opportunity to revise our manuscript. We have carefully read all your comments as well as those of the reviewers. Subsequently, we have made thorough and comprehensive revisions to our manuscript. We hope that our modifications meet the high standards of both you, the reviewers, and the journal. If you have any further comments, we would be more than willing to engage in further discussion. Below are our responses to the comments from you and the reviewers:

1.If applicable, we recommend that you deposit your laboratory protocols in protocols.io to enhance the reproducibility of your results. Protocols.io assigns your protocol its own identifier (DOI) so that it can be cited independently in the future. For instructions see: https://journals.plos.org/plosone/s/submission-guidelines#loc-laboratory-protocols. Additionally, PLOS ONE offers an option for publishing peer-reviewed Lab Protocol articles, which describe protocols hosted on protocols.io. Read more information on sharing protocols at https://plos.org/protocols?utm_medium=editorial-email&utm_source=authorletters&utm_campaign=protocols.

Response: Dear Editor, Our study is based on genome-wide association studies from existing public databases. Therefore, storing experimental protocols on Protocols.io seems inappropriate for our research. However, to ensure clarity for other readers regarding the methodology of our study, we have provided detailed descriptions of the entire study design and all data sources in the manuscript. This will enable other researchers to replicate our study smoothly based on our explanations.

Response: We have carefully reviewed the style requirements of PLOS One and have made comprehensive and meticulous revisions to the manuscript.

3.We suggest you thoroughly copyedit your manuscript for language usage, spelling, and grammar. If you do not know anyone who can help you do this, you may wish to consider employing a professional scientific editing service. 

Whilst you may use any professional scientific editing service of your choice, PLOS has partnered with both American Journal Experts (AJE) and Editage to provide discounted services to PLOS authors. Both organizations have experience helping authors meet PLOS guidelines and can provide language editing, translation, manuscript formatting, and figure formatting to ensure your manuscript meets our submission guidelines. To take advantage of our partnership with AJE, visit the AJE website (http://aje.com/go/plos) for a 15% discount off AJE services. To take advantage of our partnership with Editage, visit the Editage website (www.editage.com) and enter referral code PLOSEDIT for a 15% discount off Editage services. If the PLOS editorial team finds any language issues in text that either AJE or Editage has edited, the service provider will re-edit the text for free.

Response: Dear editor, we have carefully read all the review comments from you and the three reviewers, and have done our best to make detailed revisions to the manuscript. And the manuscript has been marked accordingly according to your comments. We hope that our revisions can meet the high standards of you and the reviewers. If you and the reviewers have more comments, we will be happy to communicate with you further. We used the professional language editing service from Bullet Edits in the UK and uploaded the service certification documents.

4.Thank you for stating in your Funding Statement: 

This study was supported by The Key Research and Development Plan (Major Science and Technology Special Project) of the Sichuan Provincial Science and Technology Department (No.2022YFS0392) and the Key Discipline Construction Project of the Sichuan Provincial Administration of Traditional Chinese Medicine (No.202072).

Response: Dear Editor, We have refined our funding information in accordance with the requirements of PLOS One and described it in the cover letter.

5.Please include captions for your Supporting Information files at the end of your manuscript, and update any in-text citations to match accordingly. Please see our Supporting Information guidelines for more information: http://journals.plos.org/plosone/s/supporting-information. 

Response: We have added supporting information to the manuscript.

Response to Reviewer #1

Dear Reviewer #1, thank you for your valuable suggestions to improve our manuscript. We have improved the manuscript based on your comments. Below are responses to your comments.

1.In Table 1, the definitions of various phenotypes used in the GWAS data should have a more detailed range based on the International Classification of Diseases.

Response: Dear Reviewer, we have provided detailed supplemental definitions for the phenotypes used in the genome-wide association study. You can refer to Table 1 in our latest submission for clarification.

2.If possible, I suggest the authors include leave-one-out results, funnel plots, and scatter plots for the mediation Mendelian randomization analysis in the supplementary methods section.

Response: Dear reviewer, we have incorporated leave-one-out, funnel plots, and scatter plots results involved in the mediation Mendelian randomization analysis into the supplementary methods. We appreciate your meticulous requirements for our manuscript, which have contributed to its improvement.

Response to Reviewer #2

Dear Reviewer #2, we sincerely appreciate your valuable feedback and admire your rigorous scientific spirit. We have carefully reviewed all of your comments and made meticulous revisions to our manuscript. It is evident that the quality of our paper has significantly improved, thanks to your insightful suggestions for modification. Through this revision process, we have gained valuable expertise in manuscript writing, which we will continue to apply and refine in our future work. Below is our detailed response to your feedback, and we hope our revisions meet your high standards. Please do not hesitate to contact us if you have any further review comments.

1. I recommend revising the title. Please present a title that conveys the most important message of this manuscript. Suggestions: "Unraveling the Causal Relationship Between Migraine and Gastroesophageal Reflux Disease: A Mendelian Randomization Study"; "Migraine and GERD: Disentangling the Complex Connection with Depression as a Mediator" [1 –3].

Response: After careful consideration of the characteristics of our study, we have chosen the second manuscript title recommended by you.

2. A graphical abstract that visually summarizes the main findings of this manuscript is highly recommended.

Response: We have created a graphical abstract based on our research findings. You can find it in the latest manuscript.

3. Abstract: I would like the authors to make as much effort in this section for the remainder of the manuscript. Please expand the abstract in 200 words (preferably 200–220 words; max. 250), although the journal allows 300 words [4], focusing on proportionally presenting the background, methods, results, and conclusion (without the headings of the subsections). The background should include the general background (one to two sentences), the specific background (two to three sentences), and "the current issue addressed to this study" (one sentence), leading to the objectives. Additionally, the methods should clarify the authors’ approach, such as the study design and variables, to solve the problem and/or make progress on the problem. The results should be close to a single sentence, thus putting the results in a more general context. The conclusion should open with one sentence describing the main result using such words as “Here we show,” which should be followed by statements such as the potential and the advance this study has provided in the field, and finally a broader perspective (two to three sentences) readily comprehensible to a scientist in any discipline [5–8].

Response: Thank you for your valuable suggestions. Upon revisiting our previous abstract, we identified several shortcomings. We have made detailed revisions to the abstract based on your feedback, which you can find in the latest manuscript.

4. Keywords: Please list as many keywords as allowed by the journal, chosen from Medical Subject Headings (MeSH) [9], and use as many keywords as possible in the title and in the first two sentences of the abstract [7,8].

Response: Based on the content of this manuscript, we have searched for additional keywords in the Medical Subject Headings and incorporated them into the title and abstract wherever possible.

5. Introduction: The authors need to fully expand this section with several paragraphs made up of about 1000 words, introducing information on the main constructs of this protocol, which should be understood by a reader in any discipline, and making it persuasive enough to put forward the main purpose of the current research the authors have conducted and the specific purpose the authors have intended by this study. I would like to encourage the authors to present the introduction starting with the general background, proceeding to the specific background, and finally, the current issue addressed to this study, leading to the objectives. These main structures should be organized in a logical and cohesive manner [10]. In this regard, the manuscript would greatly benefit from incorporating a discussion of the following topics. The following studies, but not limited to, may enhance the value of this manuscript [11–17].

Response: Dear Reviewer, thank you for your valuable feedback. We have meticulously revised the introduction based on your suggestions. We carefully reviewed the literature you recommended as well as some additional sources, significantly expanding this section. We have organized our writing more effectively and included citations from many more references, resulting in a noticeable improvement in the readability and usefulness of this part.

6. Methods: I recommend opening this section with a short introductory paragraph and citing more references to ensure the reliability and integrity of the evidence in the study design that the authors built and the methodology they have decided to apply.

Response: Thank you for your feedback. We have adjusted the description in this section to make it more concise and readable. Additionally, we have added citations to appropriate places to ensure the reliability and completeness of the methods we employed.

7. Results: I suggest presenting the independent results section and closing this section with a paragraph that puts the results into a more general context.

Response: Thank you for your suggestion. We've made adjustments to this section based on your suggestions.

8. Discussion: I would like the authors to present an independent section without subheadings with several paragraphs of about 1500 words, beginning this section with an introduction and providing a summary of the previous section. Then, I expect the authors to develop arguments clarifying the potential of this study as an extension of the previous work, the implications of the findings, how this study could facilitate future research, the ultimate goal, the challenge, the knowledge and technology necessary to achieve this goal, the statement about this field in general, and the importance of this line of research. It is particularly important to present the limitations, merits, and potential translation of this study into clinical practice [18,19].

Response: Thank you very much for your suggestions. We recognize the shortcomings in the previous discussion section. Taking your advice into full consideration, we thoroughly revised this part after reading numerous literature sources. We believe that, under your guidance, the quality of our manuscript has significantly improved.

9. Conclusion: I believe that presenting this section with 150–200 words would benefit from a single paragraph that presents some thoughtful and in-depth considerations by the authors as experts to convey the main message. The authors should make an effort to explain the theoretical implications and translational applications of their research. To understand the significance of this field, I believe it is necessary to discuss theoretical and methodological avenues in need of refinement as well as future research directions.

Response: We appreciate your insightful suggestions, and have made detailed revisions to this section according to your feedback.

10. References: Please cite more references. An article like this typically cites more than 60 to 70 references.

Overall, the manuscript contains four figures, two table, and 48 references. I believe that the merit of this manuscript lies in its contribution to the understanding of the causal relationship between GERD and migraine, with depression as a mediator. The study used a robust MR method and a large sample size to provide strong evidence supporting the notion that GERD increases the risk of migraine, while migraine does not have a causal effect on GERD. The study also highlights the pivotal role of depression as a bridge in this relationship, with depression mediating a significant proportion of the effect of GERD on migraine. The findings of this study have important implications for the treatment and management of patients with GERD and migraine, particularly those with comorbid depression. The study suggests that interventions targeting GERD may help reduce the risk of migraine and that addressing depression may be an important component of such interventions. Overall, this study provides valuable insights into the complex interplay between gastrointestinal diseases and brain disorders and may pave the way for personalized treatment approaches in the future. I hope that after careful revision, the manuscript meets the journal’s high standards for publication.

Response: Dear reviewer, thank you for your affirmation of our work. We hold the utmost respect for your diligent reviewing efforts. This revision process has been significant for us, particularly in light of the valuable suggestions you provided regarding our manuscript. Your meticulous review suggestions demonstrate your high level of professional competence, an aspect we greatly admire and aspire to emulate. We have endeavored to refine our manuscript according to your recommendations, aiming for higher standards of clarity, readability, accuracy, and scientific rigor. We hope our revisions meet the high standards of both you and the journal. Should you have further suggestions, we look forward to continued communication with you. We aspire for our work to contribute to the understanding of the causal relationships between gastroesophageal reflux disease, migraines, and depression.

Response to Reviewer #3

Dear Reviewer #3, we greatly appreciate your efforts in reviewing our manuscript. We have carefully considered all

---

## [Decision Letter · Decision Letter 1]

13 May 2024

Migraine and gastroesophageal reflux disease: disentangling the complex connection with depression as a mediator

PONE-D-23-39298R1

Dear Dr. Shen,

We’re pleased to inform you that your manuscript has been judged scientifically suitable for publication and will be formally accepted for publication once it meets all outstanding technical requirements.

Kind regards,

Y-h. Taguchi, Dr. Sci.

Academic Editor

PLOS ONE

Additional Editor Comments (optional):

accepted

Reviewers' comments:

Reviewer's Responses to Questions

**Comments to the Author**

1. If the authors have adequately addressed your comments raised in a previous round of review and you feel that this manuscript is now acceptable for publication, you may indicate that here to bypass the “Comments to the Author” section, enter your conflict of interest statement in the “Confidential to Editor” section, and submit your "Accept" recommendation.

Reviewer #1: (No Response)

Reviewer #2: All comments have been addressed

Reviewer #3: All comments have been addressed

2. Is the manuscript technically sound, and do the data support the conclusions?

Reviewer #1: Yes

Reviewer #2: Yes

Reviewer #3: Yes

3. Has the statistical analysis been performed appropriately and rigorously? 

Reviewer #1: Yes

Reviewer #2: Yes

Reviewer #3: Yes

4. Have the authors made all data underlying the findings in their manuscript fully available?

Reviewer #1: Yes

Reviewer #2: Yes

Reviewer #3: Yes

5. Is the manuscript presented in an intelligible fashion and written in standard English?

Reviewer #1: Yes

Reviewer #2: Yes

Reviewer #3: Yes

6. Review Comments to the Author

Reviewer #1: My concerns have been addressed by the authors and it can be accepted for publication at the current version.

Reviewer #2: 6 May 2024

The 2nd review on the manuscript, titled ‘Gastroesophageal reflux disease increases migraine risk with depression as an important mediator: A Mendelian randomization study’ by Shen Z et al., submitted to Plos One

Manuscript ID: PONE-D-23-39298R1

Dear Authors,

I am pleased that the authors have addressed the issues raised in the previous round. Currently, the manuscript is a well-written research paper with informative layouts that presents used the mendelian randomization method to assess the causal relationship between migraine and gastroesophageal reflux disease. I believe the manuscript meets the journal’s high standards for publication. I am looking forward to seeing more papers written by the same authors.

Thank you!

I declare no conflict of interest regarding this manuscript.

Best regards,

Reviewer

Reviewer #3: (No Response)

7. PLOS authors have the option to publish the peer review history of their article (what does this mean?). If published, this will include your full peer review and any attached files.

Reviewer #1: No

Reviewer #2: No

Reviewer #3: No

---

## [Editor Report · Acceptance letter]

21 May 2024

PONE-D-23-39298R1 

PLOS ONE

Dear Dr. Shen, 

I'm pleased to inform you that your manuscript has been deemed suitable for publication in PLOS ONE. Congratulations! Your manuscript is now being handed over to our production team.

Kind regards, 

on behalf of

Professor Y-h. Taguchi 

Academic Editor

PLOS ONE